# Novel Cluster AZ *Arthrobacter* phages Powerpuff, Lego, and YesChef exhibit close functional relationships with *Microbacterium* phages

**Andrew Kapinos, Pauline Aghamalian, Erika Capehart, Anya Alag, Heather Angel, Eddie Briseno, Byron Corado Perez, Emily Farag, Hilory Foster, Abbas Hakim, Daisy Hernandez-Casas, Calvin Huang, Derek Lam, Maya Mendez, Ashley Min, Nikki Nguyen, Alexa L. Omholt, Emily Ortiz, Lizbeth Shelly Saldivar, Jack Arthur Shannon, Rachel Smith, Mihika V. Sridhar [ORCID], An Ta [ORCID], Malavika C. Theophilus, Ryan Ngo, Canela Torres, Krisanavane Reddi, Amanda C. Freise, Jordan Moberg Parker [ORCID]¤***

Department of Microbiology, Immunology and Molecular Genetics, University of California, Los Angeles, CA, United States of America

¤ Current address: Department of Biomedical Science, Kaiser Permanente Bernard J. Tyson School of Medicine, Pasadena, CA, United States of America

\* jordan.p.parker@kp.org

**Data Availability Statement:** All genome sequence files are available from the GenBank database

## Abstract

Bacteriophages exhibit a vast spectrum of relatedness and there is increasing evidence of close genomic relationships independent of host genus. The variability in phage similarity at the nucleotide, amino acid, and gene content levels confounds attempts at quantifying phage relatedness, especially as more novel phages are isolated. This study describes three highly similar novel *Arthrobacter globiformis* phages–Powerpuff, Lego, and YesChef– which were assigned to Cluster AZ using a nucleotide-based clustering parameter. Phages in Cluster AZ, *Microbacterium* Cluster EH, and the former *Microbacterium* singleton Zeta1847 exhibited low nucleotide similarity. However, their gene content similarity was in excess of the recently adopted *Microbacterium* clustering parameter, which ultimately resulted in the reassignment of Zeta1847 to Cluster EH. This finding further highlights the importance of using multiple metrics to capture phage relatedness. Additionally, Clusters AZ and EH phages encode a shared integrase indicative of a lysogenic life cycle. In the first experimental verification of a Cluster AZ phage's life cycle, we show that phage Powerpuff is a true temperate phage. It forms stable lysogens that exhibit immunity to superinfection by related phages, despite lacking identifiable repressors typically required for lysogenic maintenance and superinfection immunity. The ability of phage Powerpuff to undergo and maintain lysogeny suggests that other closely related phages may be temperate as well. Our findings provide additional evidence of significant shared phage genomic content spanning multiple actinobacterial host genera and demonstrate the continued need for verification and characterization of life cycles in newly isolated phages.

(accession numbers MN703413, MT024869, MT024871).

**Funding:** The authors received no specific funding for this work.

**Competing interests:** The authors have declared that no competing interests exist.

## Introduction

Bacteriophages comprise the most abundant group of biological entities on the planet, with an estimated $10^{31}$ phage particles in existence [1]. There is a growing body of evidence suggesting the immense role phages play in ecological regulation through interactions with their bacterial hosts [2–4]. Despite this the global phage population remains relatively understudied, with only 3,758 actinobacteriophage genomes published to PhagesDB as of August 2021 [5].

Actinobacteriophages display immense genomic and biological diversity [6]. Past studies have observed that phages infecting the same bacterial host and exhibiting the same viral life cycle tend to share the highest amount of nucleotide similarity, with a more conserved evolutionary history [7]. However, substantial levels of genomic diversity have been identified even amongst phages known to infect a common host [6]. Studies of phage relatedness are further complicated by the mosaic nature of phage genomes, due to widespread exchanges of modules of genetic material [8]. Given that host barriers to genetic exchange are more readily violable than previously thought, this can result in phages of unique bacterial hosts sharing considerable gene content [9]. A recent study of a large collection of *Microbacterium* phages described significant shared gene content amongst a group of phages infecting *Microbacterium*, *Streptomyces*, *Rhodococcus*, *Gordonia*, and *Arthrobacter* spp. [10]. It was also found that sequenced *Microbacterium* phages exhibited shared gene content or genome architecture with *Arthrobacter* phages. This evidence suggested that phages infecting *Microbacterium* and *Arthrobacter* spp. may exhibit proximal phylogenetic relationships.

Few studies have specifically explored the phages that infect *Arthrobacter*, a genus of bacteria that is primarily soil-dwelling and engaged in the biochemical processing of natural compounds [11–13]. Klyczek *et al.* described a collection of *Arthrobacter* phages, all isolated on *Arth*robacter *sp*. ATCC 21022, which shared no nucleotide sequence similarity with phages infecting other actinobacterial hosts [14]. These *Arthrobacter* phages were considered to be primarily lytic, similar to sequenced *Microbacterium* phages [10], and unlike sequenced *Mycobacterium* and *Gordonia* phages which are more likely to be temperate [9, 15]. Of the 331 sequenced *Arthrobacter* phages on PhagesDB as of August 2021, only 61 are predicted to be temperate, comprising Clusters AS, AY, AZ, FA, FF, and FG [5]. Importantly, many predictions of *Arthrobacter* phage life cycles have depended on bioinformatic evidence, such as the presence of a known integrase, and have yet to be verified experimentally. Additional analyses of potential genomic relationships of *Arthrobacter* phages, including investigations of amino acid identity and Gene Content Similarity (GCS) [9], have so far been limited in scope. The isolation of novel *Arthrobacter* phages allows for more thorough genomic comparisons to phages infecting both *Arthrobacter* and other actinobacterial hosts and provides the opportunity for experimental verification of phage life cycles.

The first goal of this study was to describe the relationships of novel *Arthrobacter globiformis* phages Powerpuff, Lego, and YesChef to phages infecting *Arthrobacter* and non-*Arthrobacter* hosts. These phages were determined to be members of the actinobacteriophage Cluster AZ using a nucleotide-based clustering parameter [14, 16]. We discovered that, while Cluster AZ phages shared minimal similarity with *Microbacterium* and *Streptomyces* phages at the nucleotide level, phages in Cluster AZ shared GCS with Microbacterium Cluster EH phages in excess of the recently adopted 35% clustering threshold [10]. The *Microbacterium* phage Zeta1847, previously designated as a singleton phage, was also found to share over 35% GCS with all Cluster EH phages analyzed and was reassigned to Cluster EH. The second goal of this paper was to experimentally verify a Cluster AZ phage's life cycle. We found that the novel phage Powerpuff is a true temperate phage likely encoding a repressor that has yet to be

identified. In sum, we present the first comparative genomic study of phages belonging to actinobacteriophage Cluster AZ and verify, for the first time, the life cycle of a phage in this cluster.

## Methods

### Phage isolation, purification, and amplification

Three soil samples were collected from within Los Angeles County, CA, USA: 34.443624 N, 118.609545 W (Powerpuff), 34.016253 N, 118.501056 W (Lego), and 34.052707 N, 118.44657 W (YesChef). Direct isolation of phage YesChef was performed at 30˚C using PYCa broth (Yeast Extract 1 g/L, Peptone 15 g/L, 4.5mM CaCl$_2$, Dextrose 0.1%), while enriched isolations of phages Powerpuff and Lego were performed at 25˚C and 30˚C, respectively, using 10X PYCa broth (Yeast Extract 10 g/L, Peptone 150 g/L, 45mM CaCl$_2$, Dextrose 10%) and *Arthrobacter globiformis* B-2979. Filter-sterilized samples were spot tested using *A. globiformis* B-2979 and PYCa media using the double agar overlay method. Samples containing putative phage were purified and amplified as described previously [17].

### Transmission Electron Microscopy (TEM)

Each high titer lysate was aliquoted onto a carbon-coated grid and stained using 1% (w/v) uranyl acetate. Each carbon grid was imaged using a FEI T12 TEM Instrument (Thermo Fisher Scientific, MA, USA) at magnifications between 30,000X and 42,000X. Phage capsid and tail measurements were determined using ImageJ [18].

### DNA extraction and sequencing

DNA was extracted from high titer lysates using the Wizard® Clean-Up Kit (cat. # A7280, Promega, WI, USA). Sequencing libraries were constructed with the NEBNext® UltraTM II DNA Library Prep kit (New England Biolabs, MA, USA), and shotgun sequenced by Illumina-MiSeq at the Pittsburgh Bacteriophage Institute. Genome assembly and finishing were conducted as previously described [19].

### Genome annotation

The Phage Evidence Collection and Annotation Network (PECAAN) was used to document evidence during manual annotation of phage genomes (https://discover.kbrinsgd.org/). Genes were preliminarily auto-annotated using DNA Master (http://cobamide2.bio.pitt.edu). Gene-Mark [20] and Glimmer [21] were used to assess coding potential. Phamerator was used to assign genes to phamilies (phams) on the basis of amino acid similarity and to examine synteny with related phages [22]. Conserved start sites were identified using Starterator [23]. PhagesDB BLASTp [5], NCBI BLASTp [24], the NCBI Conserved Domain Database [25], and HHpred [26] were used for gene function calls. Membrane protein topology programs TmHmm [27] and TOPCONS [28] were used to identify putative transmembrane domains within draft genes.

### Comparative genomic analyses

Upon the completion of manual annotation, the final version of each phage genome was downloaded from Phamerator and used to create a linear genome map using Inkscape 1.0 (https://inkscape.org/). NCBI Nucleotide BLAST (BLASTn) was optimized for highly similar sequences (megablast) and used to identify similar phage genomes. Gepard 1.40 was used to generate dotplots using word sizes of 15 and 5 for nucleotide and amino acid inputs,

respectively [29]. OrthoANIu and coverage values were calculated using the command-line OrthoANIu tool provided by EZBioCloud [30] and visualized as a heat map using Prism 8.0.0 (Graphpad Software, San Diego, California, USA).

Pham data for phages of interest from the Actino_Draft database (version 382) were input into SplitsTree 4.16.1 to produce a network phylogeny using default parameters [31]. SplitsTree uses the presence or absence of each gene to construct a "network" that represents such differences visually and qualitatively. Gene Content Similarity between phages of interest was calculated using the PhagesDB Explore Gene Content tool [5] and visualized as a heatmap using Graphpad Prism 8.0.0. Specific information regarding pham presence and function in each phage of interest was collected using PhagesDB and Phamerator.

### Host range assay

For each undiluted high titer lysate (>5x109 PFU/mL) of phages Powerpuff, Lego, and YesChef, 3uL were spotted onto a plate spread with a lawn of *Microbacterium foliorum* NRRL B-24224 on PYCa media. Plates were incubated at 25˚C for 48 hours and examined for clearings (plaques). As a positive control, the lysates were also spotted onto a lawn of *A. globiformis* B-2979 on PYCa media plates, then incubated at 25˚C for 48 hours and examined for clearings.

### Preparation of stable lysogens and immunity assays

Powerpuff high titer lysate was serially diluted and spotted onto *A. globiformis* B-2979, then incubated at 30˚C for 96 hours. All subsequent immunity assay plates were incubated at 30˚C for 48 hours. Bacterial mesas from spot dilutions $10^0$ through $10^{-3}$ were streak purified three times on PYCa media to remove exogenous phage particles. Experimental plates were prepared by streaking putative lysogens onto a prepared lawn of *A. globiformis* B-2979 (patch test) to assay for the release of phage, while control plates were prepared in the absence of host cells to confirm lysogen viability.

A liquid release assay was performed to verify the presence of stable lysogens and assay for the release of lytically active phage. Liquid cultures of streak purified putative lysogens were incubated at 30˚C for 48 hours and then pelleted to remove bacterial cells. Ten-fold serial dilutions of supernatants containing putative released phages were spot tested on *A. globiformis* B-2979 to confirm phage release and calculate titer. Immunity assays of *Arthrobacter* Cluster AZ phages Powerpuff, Lego, and YesChef, Cluster FE phage BlueFeather, Cluster AU phage Giantsbane, and Cluster AO phage Abba were performed using ten-fold serial dilutions of phage lysate on wild-type (WT) *A. globiformis* B-2979 and *A. globiformis* B-2979 lysogens of Powerpuff.

## Results

### Phages Powerpuff, YesChef, and Lego are highly similar *Siphoviridae* members of Cluster AZ

Phages Powerpuff, Lego, and YesChef all exhibited 1–3 mm turbid bullseye plaques after 24 hours of incubation at their respective isolation temperatures (Fig 1). Transmission electron microscopy of the three phages revealed similar particle dimensions, with an average head diameter of 56.9 nm and an average tail length of 124.5 nm (Table 1). All phages exhibited long, flexible tails, indicative of *Siphoviridae* [32].

Genome sequencing and assembly determined that all three phages exhibited 11 base 3' sticky overhangs (CGAAGGGGCAT), with similar genome length, percent GC content, and number of genes (Table 1). Powerpuff, Lego, and YesChef were assigned to Cluster AZ using a

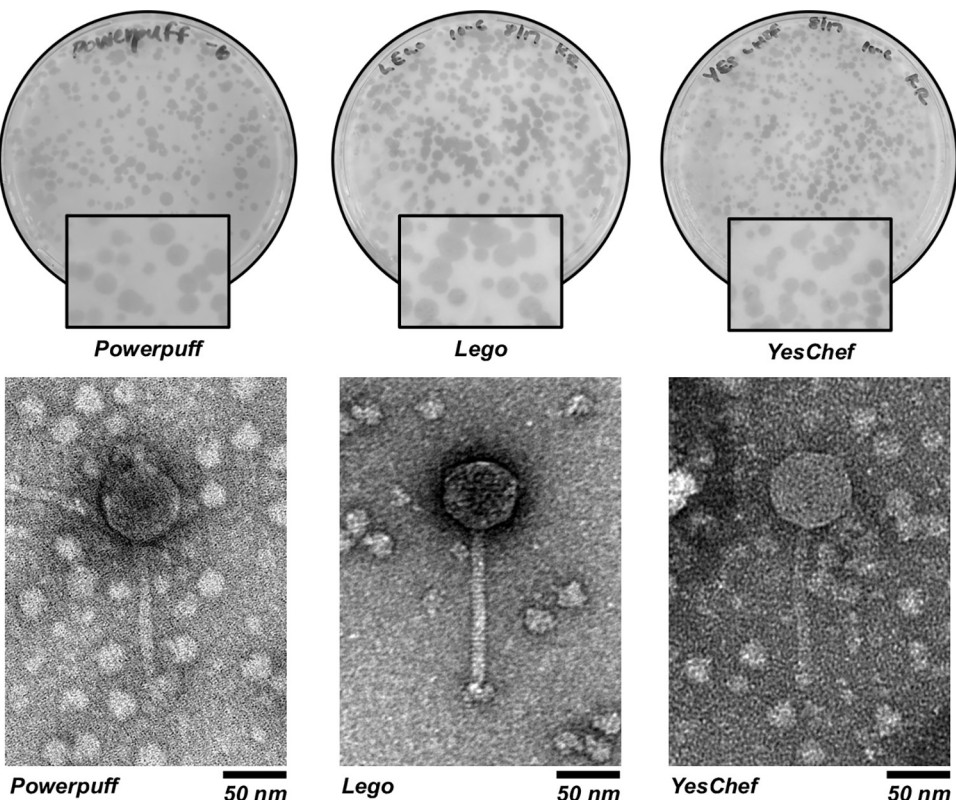

**Fig 1. Phages Powerpuff, Lego, and YesChef exhibit turbid bullseye plaque morphologies and are *Siphoviridae*.**
Purified phage lysates were plated using the double agar overlay method. Each phage exhibited 1–3 mm turbid bullseye plaques, suggestive of a lysogenic life cycle. The presence of long, flexible tails supported their classification as *Siphoviridae*.

nucleotide-based parameter [14, 16]. Powerpuff, Lego, and YesChef were also found to be nearly genomically identical according to a preliminary BLASTn search, which revealed pairwise BLASTn coverages of at least 99%, with E-values of 0 and identities of at least 98.63% (S1 Table). These nucleotide similarities translated into high similarity in gene content and genome architecture, with few differences between these phages at the gene level (S1 Fig). Only Cluster AZ phages Tbone and Kaylissa shared between 96.26–96.86% BLASTn identity and 92–98% coverage with Powerpuff, Lego, and YesChef, with all pairwise comparisons having E-values of 0. The remaining 7 phages analyzed from Cluster AZ shared between 80.14–87.79% BLASTn identity and 8–81% coverage with the novel phages. While Powerpuff, Lego, and YesChef were isolated from within Los Angeles County, phages Tbone and Kaylissa were isolated from Louisiana and New York state, respectively [5]. This provides another interesting case of phages which are extremely similar genomically, despite being isolated from locations which are geographically distant [33].

**Table 1. Phages Powerpuff, Lego, and YesChef share similar genomic and physical characteristics.**

| Phage | Accession | Genome length (bp) | %GC Content | No. of genes | Head diam. (nm) | Tail length (nm) |
|---|---|---|---|---|---|---|
| Powerpuff | MN703413 | 44651 | 67.6% | 71 | 56.7 ± 13.9 | 126.7 ± 17.0 |
| Lego | MT024869 | 43446 | 67.5% | 69 | 57.6 ± 2.1 | 120.4 ± 10.7 |
| YesChef | MT024871 | 43510 | 67.7% | 69 | 56.5 ± 5.2 | 126.3 ± 1.9 |

## Cluster AZ *Arthrobacter* phages are diverse and share nucleotide similarity with Cluster EH *Microbacterium* phages

Nucleotide comparisons of Powerpuff, Lego, and YesChef to the most genomically similar actinobacteriophages both within and outside of Cluster AZ were performed. It was expected that the most similar set of phages to Powerpuff, Lego, and YesChef at the nucleotide level would also be members of *Arthrobacter*-infecting Cluster AZ, which was confirmed using PhagesDB BLASTn (S2 Table). The most similar phages to Powerpuff, Lego, and YesChef outside of Cluster AZ included phages in *Microbacterium* Clusters EH and EB, *Arthrobacter* Cluster AK, and *Streptomyces* Cluster BJ. Phage and cluster information is available online at phagesDB.org (S3 Table, obtained November 2021) [5].

Nucleotide dotplot comparisons using a word length of 15 revealed that Cluster AZ phages Liebe and Maureen exhibited strong alignments to one another but weak alignments to the remainder of their cluster (Fig 2). As expected from the BLASTn results, Cluster AZ shared some degree of nucleotide similarity with *Microbacterium* Cluster EH phages, and significantly less similarity with Cluster EB phages. There were no nucleotide alignments observed between Cluster AZ and phages in Clusters AK or BJ.

Similarities observed at the nucleotide level were confirmed using OrthoANIu (Fig 3), which quantifies the similarity in orthologous nucleotide sequences between genomes [34]. Intracluster OrthoANIu values tended to be high (at or above 70%), well in excess of the 50% identity threshold required for clustering under nucleotide-based parameters [16]. It is notable that high intercluster OrthoANIu values also existed, such as those between Clusters AK, AZ, and EB phages. The vast majority of such comparisons exhibited coverage values below 5%.

As expected from the weak intracluster nucleotide dotplot alignments of Liebe and Maureen to the other Cluster AZ phages (Fig 2), comparisons to Liebe and Maureen accounted for the lowest OrthoANIu and coverage values both within Cluster AZ and between Clusters AZ and EH (Fig 3). The most similar phages from between Clusters AZ and EH exhibited higher OrthoANIu and coverage values than the most dissimilar phages within Cluster AZ. This was confirmed using PhagesDB BLASTn, in which the score and E-value of the weakest comparison between two Cluster AZ phages (Liebe v. Adolin; 313 bits score, E-value $3e^{-82}$) were lower than the strongest comparison between a Cluster AZ and Cluster EH phage (Yang v. IAmG-root/GardenState; 389 bits score, E-value $1e^{-105}$). This finding highlights the diversity of the Cluster AZ phages, as well as the proximity of Clusters AZ and EH in the genetic landscape.

## Amino acid sequences are similar between *Arthrobacter*, *Microbacterium*, and *Streptomyces* phages

Codon degeneracy allows for phage sequences to be shared at the amino acid level but not at the nucleotide level [9, 35], which may limit the apparent similarity of phage genomes when only comparing nucleotide sequences. Amino acid dotplot comparisons revealed similarity between *Arthrobacter* phage Clusters AZ and AK, despite minimal nucleotide identity (Fig 4). Within Cluster AZ, Liebe and Maureen exhibited strong alignments to the remainder of their cluster, in contrast with the nucleotide similarity results. Amino acid similarities between Cluster AZ and *Microbacterium* phage Clusters EH and EB, as well as *Streptomyces* phages belonging to Cluster BJ, were also stronger than nucleotide similarities. This increase in alignment at the amino acid level is indicative of synonymous substitutions in the nucleotide code, perhaps suggesting a distant evolutionary relationship for alignments which are strengthened or apparent only at the amino acid level [36].

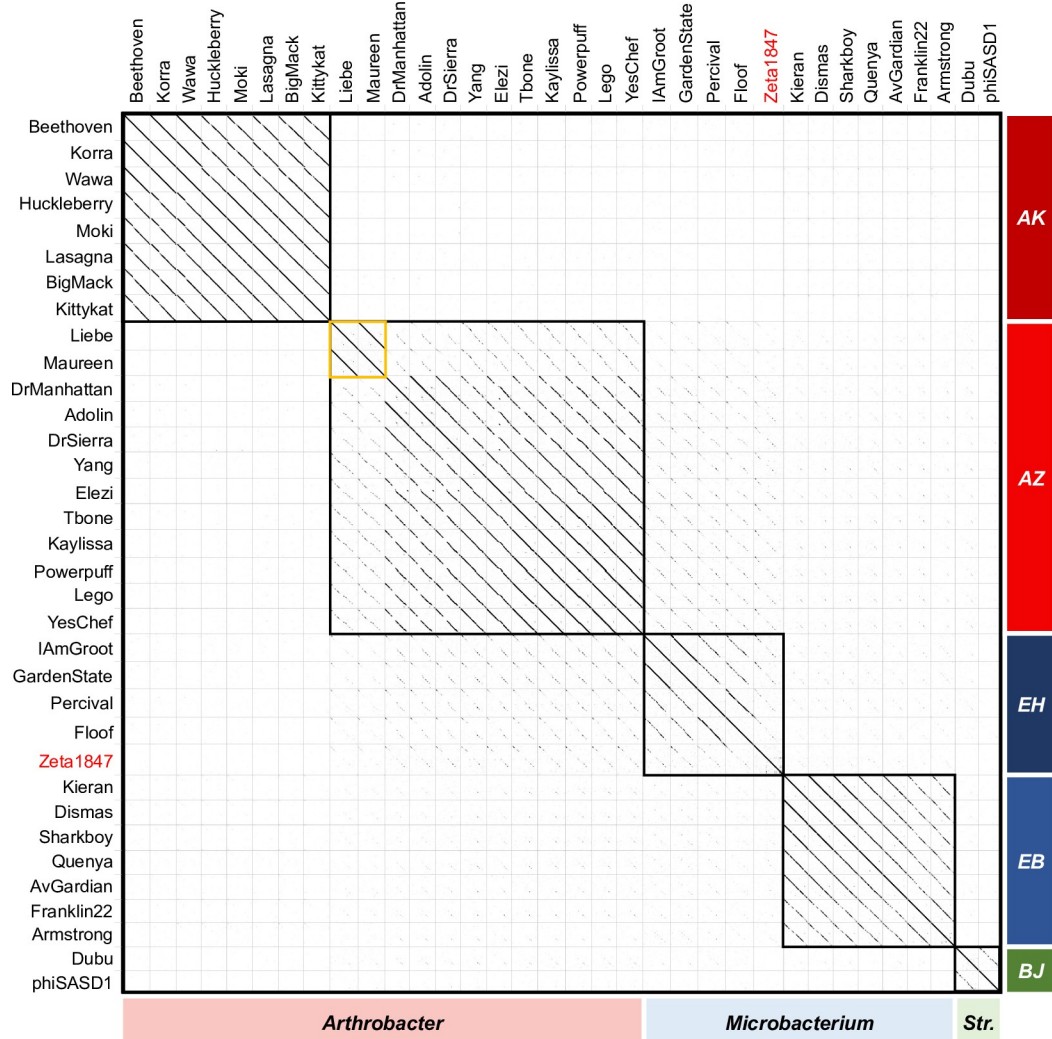

**Fig 2. Nucleotide dotplots reveal two groups of highly similar phages within Cluster AZ, with some similarity to** *Microbacterium* **phages sampled.** Whole genome nucleotide sequences were analyzed using Gepard software and a word size of 15. Cells boxed in black represent phage clusters. The former singleton Zeta1847 is indicated in red. "*Str.*" indicates phages infecting *Streptomyces*. Within Cluster AZ, phages Liebe and Maureen exhibited strong alignments to each other but weak alignments to the remainder of their cluster (indicated in yellow).

### Clusters AZ and EH share genome architecture and gene content in excess of *Microbacterium* clustering parameters

Numerous phages have been observed to share substantial portions of their gene content despite lacking significant nucleotide similarity and/or span-length coverage [9, 35]. This has prompted the adjustment of clustering parameters for new phage clusters from a nucleotide-based parameter [16] to an updated threshold of at least 35% shared gene content [9, 10]. Thus, while nucleotide and amino acid comparisons serve as important preliminary metrics for determining similarity between phages, analyses of shared gene content may serve as more functionally relevant metrics for phage comparison.

 GCS values were calculated for each genome pair included in the nucleotide and amino acid comparisons performed above. Interestingly, the putative singleton phage Zeta1847 displayed GCS values with Cluster EH phages in excess of the *Microbacterium* phage clustering

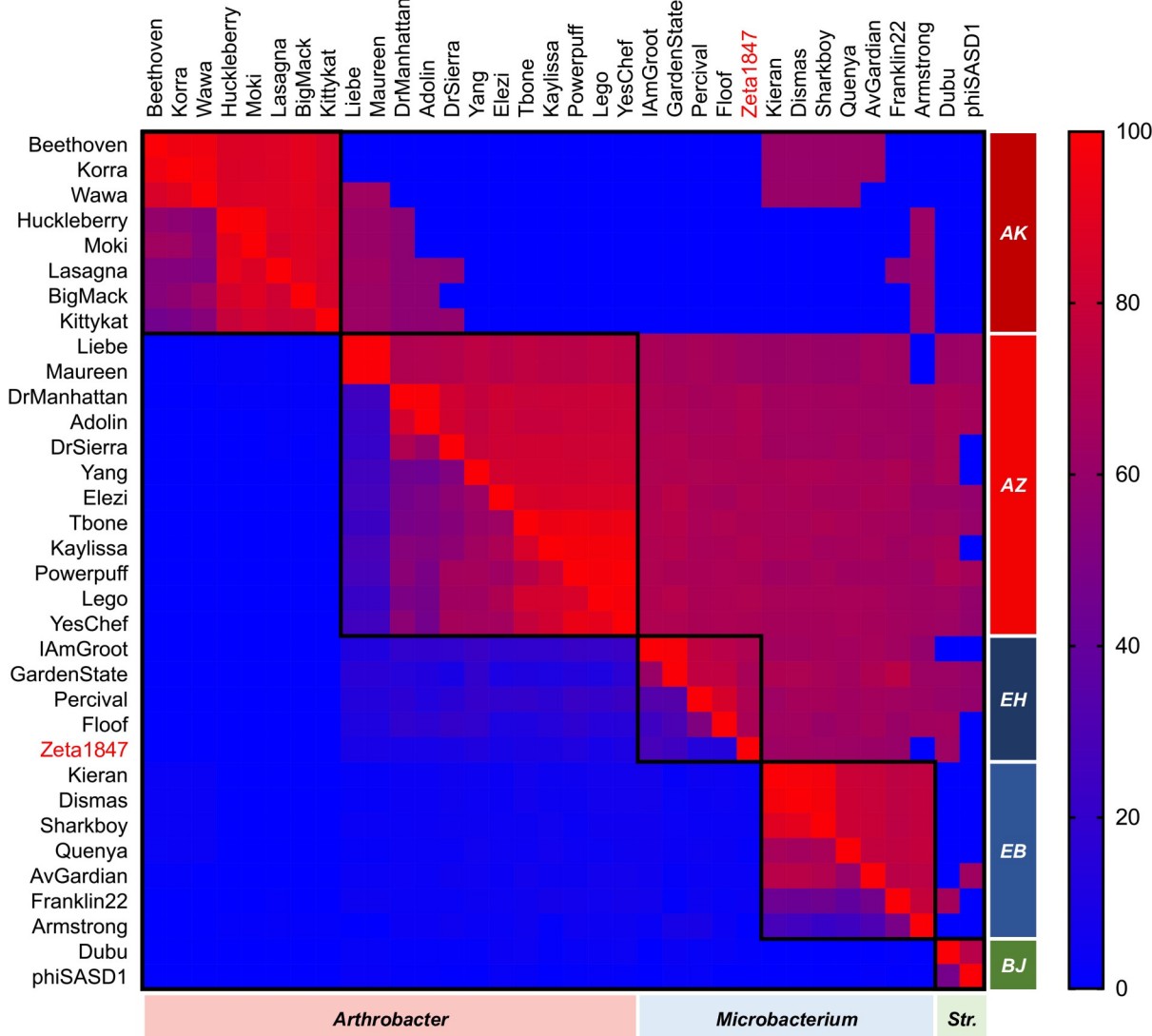

**Fig 3. OrthoANIu indicates widespread shared genomic features between Cluster AZ and *Microbacterium* phages.** Pairwise average nucleotide identities between orthologous regions of each genome (OrthoANIu; upper right values) and respective coverages (bottom left values) were calculated using a command-line OrthoANIu tool. Values were visualized as a heat map using Prism 8.0.0. Cells boxed in black represent phage clusters. The former singleton Zeta1847 is indicated in red. "*Str.*" indicates phages infecting *Streptomyces*. OrthoANIu values supported findings of the nucleotide dotplot and indicated widespread presence of small but well-conserved genomic features.

parameter, sharing between 37.5% and 40% GCS with Cluster EH phages. This finding resulted in the assignment of Zeta1847 into Cluster EH by phagesdb.org (S3 Table).

There were multiple pairwise comparisons between phages in *Arthrobacter* Cluster AZ and *Microbacterium* Cluster EH which met or exceeded the 35% GCS clustering parameter, as indicated by a white outline on the GCS heatmap (Fig 5). Cluster AZ phages DrManhattan and Adolin shared between 35.2% and 37.3% GCS with Cluster EH phages IAmGroot, GardenState, and Percival. Cluster AZ phages DrSierra and Yang also shared 35.8% and 35.3% GCS with phage Percival, respectively, while phages Liebe and Maureen shared 36.4% GCS with Cluster EH phage Floof. These comparisons appear to be as substantial as other recently identified relationships between *Arthrobacter* and *Microbacterium* phages, which also shared up to 40% GCS [10].

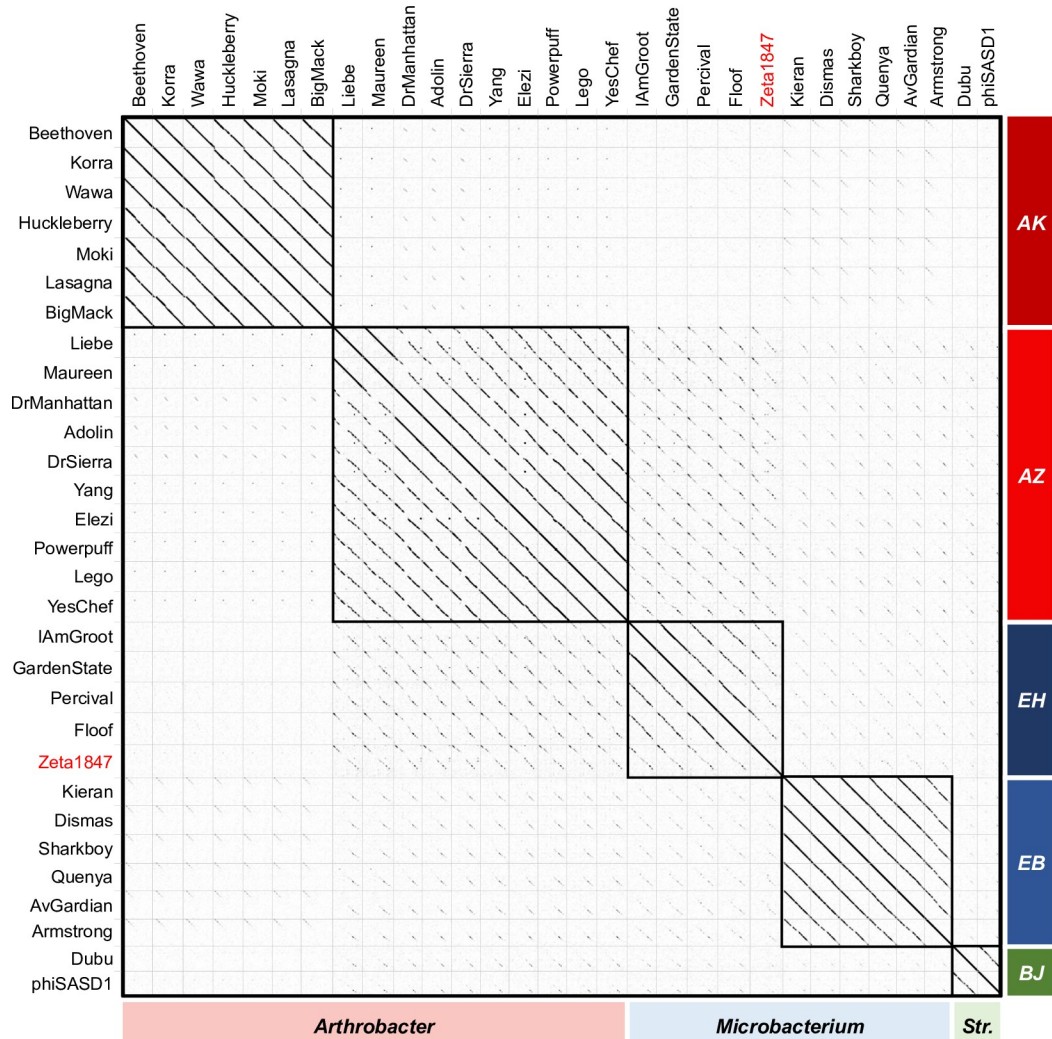

**Fig 4. Amino acid dotplots reveal a history of synonymous substitutions.** Whole genome amino acid sequences were analyzed using Gepard software and a word size of 5. Cells boxed in black represent phage clusters. The former singleton Zeta1847 is indicated in red. "*Str.*" indicates phages infecting *Streptomyces*. Increased alignment strength at the amino acid level indicated a history of synonymous substitutions and suggested distant relationships.

Phages in Clusters AZ and EH also shared similar genome architecture, with a high degree of synteny in the left arm of the genome (Fig 6). The right arm of the genome displayed less synteny between these phages; however, genes found in the same gene phamilies (phams) tended to be arranged in the same order. While Cluster AZ representative phage Powerpuff encoded an endolysin in the right arm of the genome, the Cluster EH phages encoded endolysins in the left arm. Within Cluster AZ, only phages Elezi, Liebe, and Maureen also encoded endolysins in the left arm as the Cluster EH phages do. The relative proximity of the relationships between Clusters AZ and EH was further evidenced by a SplitsTree network phylogeny of shared gene content. The goal of this experiment was to provide a qualitative visualization of group membership, which complements the quantitative data obtained by GCS analyses. Clusters AZ and EH formed a large branch separate from the rest of the phages and *Arthrobacter* Clusters AK and AZ were also segregated from one another on the tree (Fig 7). This

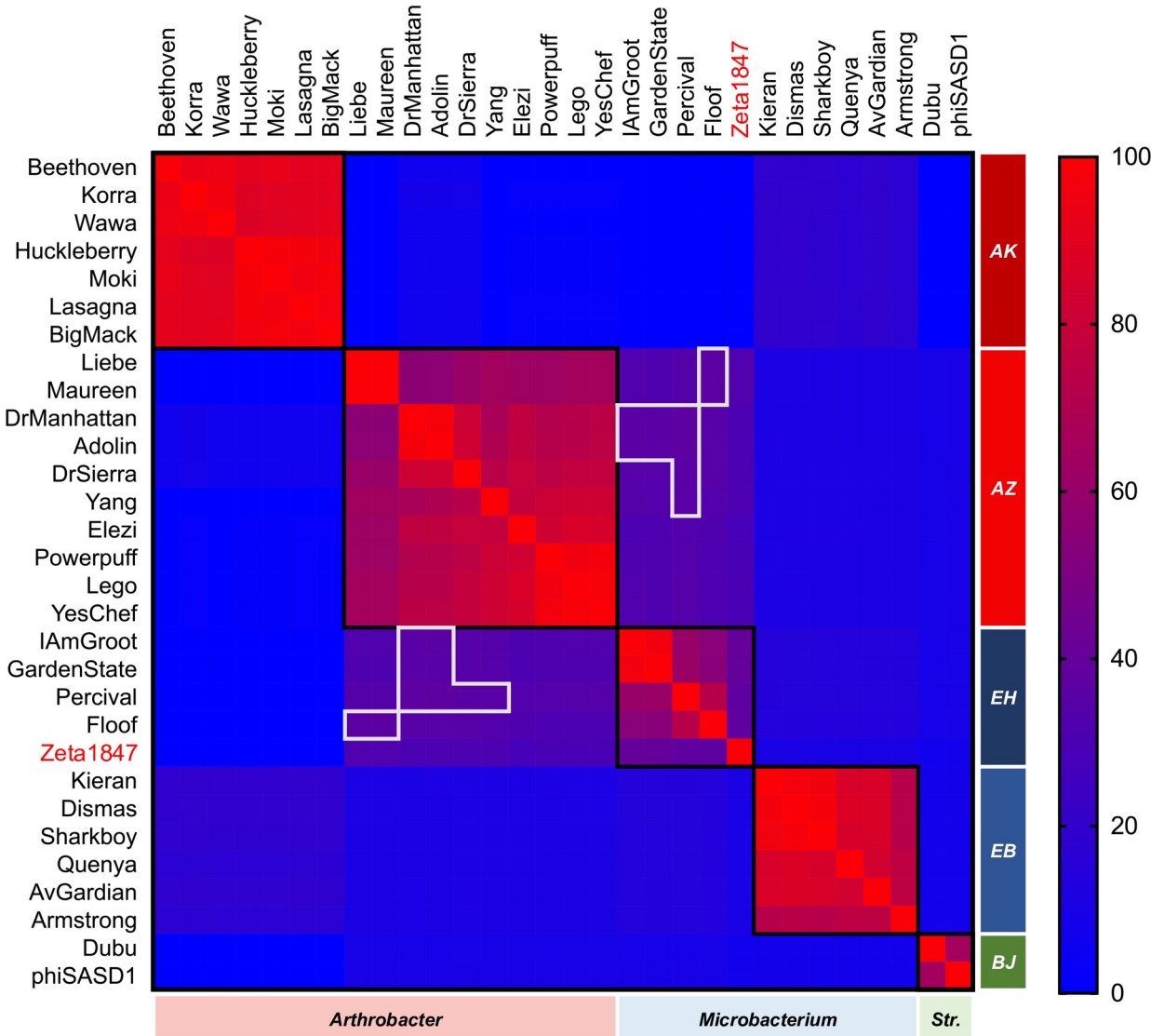

**Fig 5. Cluster AZ and EH phages share gene content in excess of *Microbacterium* clustering parameters.** GCS values were recorded using the PhagesDB Explore Gene Content tool and visualized as a heat map using Prism 8.0.0. Cells boxed in white represent pairwise GCS values in excess of gene content clustering parameters (≥35%) between phages belonging to different clusters. Cells boxed in black represent phage clusters. The former singleton Zeta1847 is indicated in red. "*Str.*" indicates phages infecting *Streptomyces*. Some phages in Clusters AZ and EH shared over 35% GCS, in excess of the *Microbacterium* clustering parameter. The former singleton Zeta1847 shared over 35% GCS with Cluster EH phages, which resulted in the clustering of this phage with Cluster EH.

supports the notion that there exists great diversity even amongst phages infecting the same host, as well as similarities between phages that have different hosts [6].

## Clusters AZ and EH phages encode for integrase and are likely temperate

An examination of the genes shared between Clusters AZ and EH revealed common functional biological features. Many DNA processing genes were shared, including the genes encoding both terminase subunits, Holliday junction resolvase, DNA polymerase I, DNA primase/helicase, and SprT-like protease. Many structural and virion assembly genes were also shared, including those encoding the portal protein, major capsid protein, head-to-tail adaptor, head-

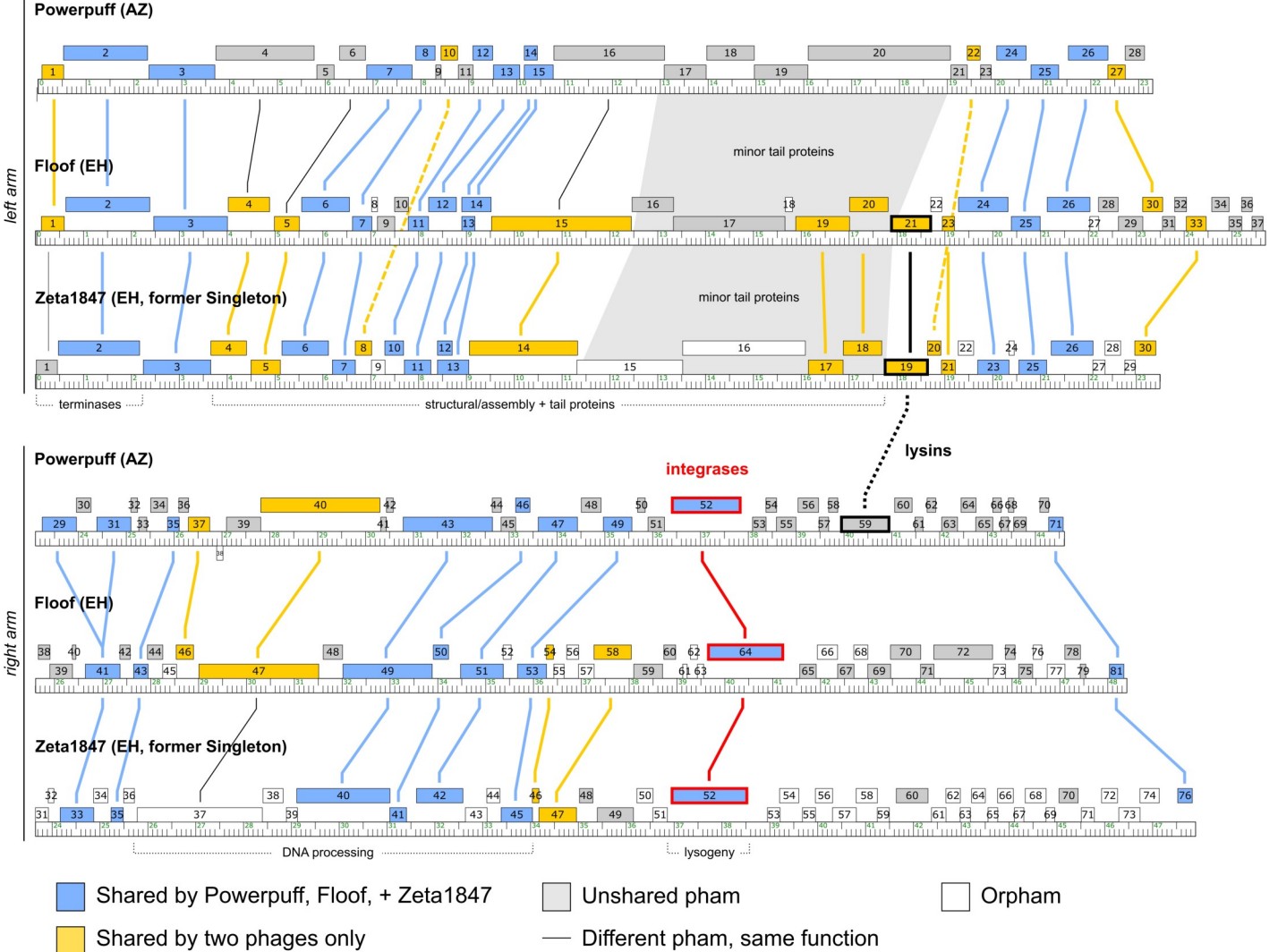

**Fig 6. Clusters AZ and EH phages share similar genome architectures.** Genome maps were downloaded from Phamerator and formatted using Inkscape 1.0. Genes in different phams with conserved functions are indicated by thin black lines and shaded regions. Integrases are highlighted in red, while lysins are highlighted in black. The left arm (top panel) of each genome was highly similar, with a less conserved right arm (bottom panel). Genes belonging to the same phams exhibited a conserved order. In Powerpuff, endolysin was found in the right arm rather than the left arm. Only Cluster AZ phages Elezi, Liebe, and Maureen encode endolysins in the left arm as do the Cluster EH phages.

to-tail stopper, tail terminator, major tail protein, and tail assembly chaperone (S1 Fig). These genes all encode for vital proteins involved in the phage life cycle and imply common biological features between these phages [9]. Given this implication, we tested the lytic activity of phages Powerpuff, Lego, and YesChef on *Microbacterium foliorum*, the isolation host of Cluster EH phages Percival and Floof. Despite sharing considerable gene content with phages infecting *M. foliorum*, Powerpuff, Lego, and YesChef were unable to infect this host (data not shown).

All phages studied in Clusters AZ and EH, except Cluster EH phage Percival, shared a pham encoding a serine integrase. Cluster EH phage Percival also encoded a serine integrase assigned to a different pham. BLASTp alignment of these two phams (Floof_64 v. Percival_59) revealed 24.89% sequence identity over 87% query coverage, with an E-value of 2e$^{-07}$,

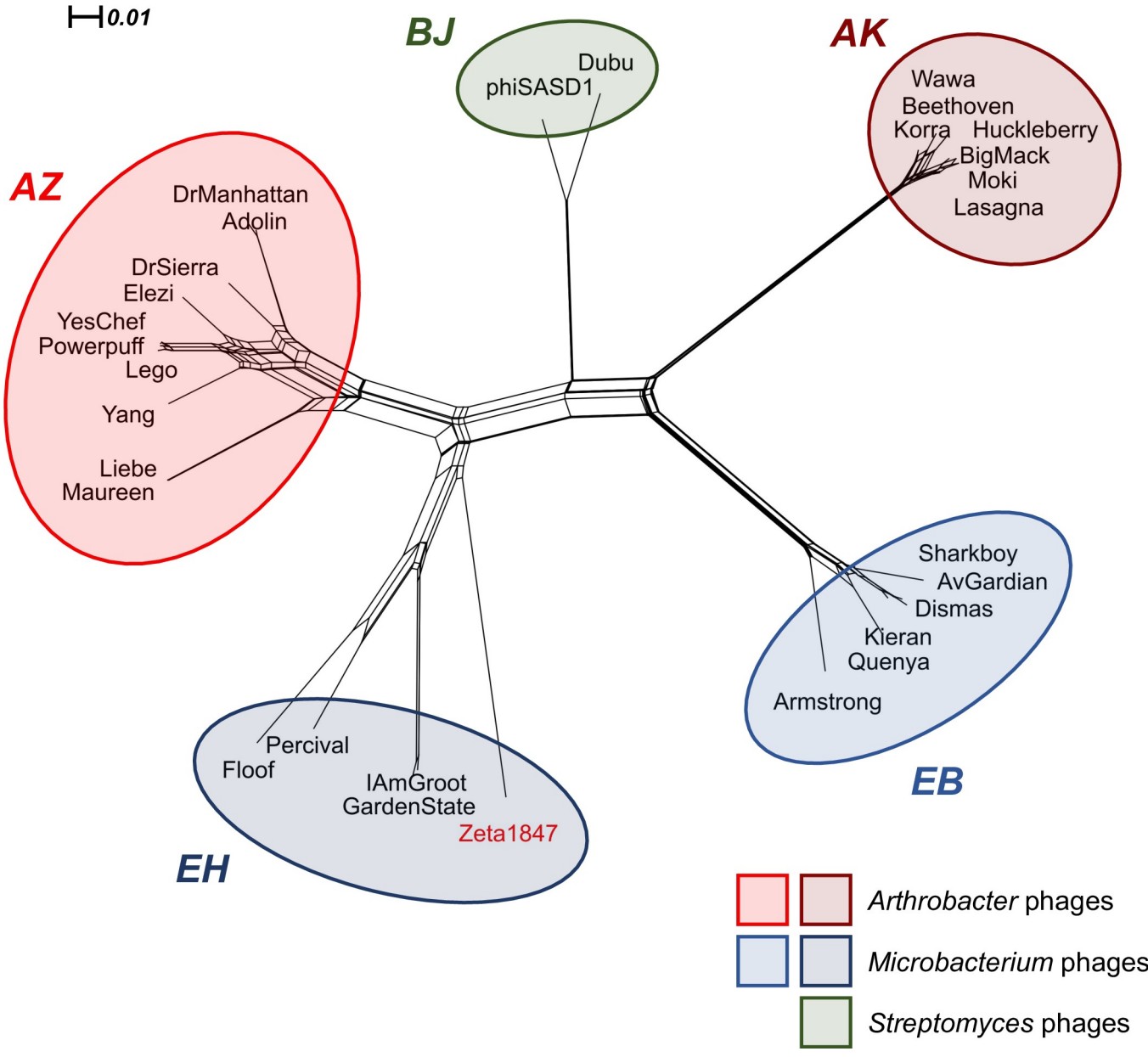

**Fig 7. Cluster AZ is more similar to *Microbacterium* and *Streptomyces* phages than to any other *Arthrobacter* phages.** Pham information was obtained from the Actino_Draft database (version 382) and input into SplitsTree 4.16.1 to produce a network phylogeny using default parameters. Phage clusters are colored by host as indicated in the legend. The former singleton Zeta1847 is indicated in red. Each cluster formed a distinct branch on the tree. The most distant group of phages from Cluster AZ comprised Cluster AK, indicating low gene similarity despite being the most closely related *Arthrobacter* phages according to BLASTn.

suggesting a distant relationship. While predicted to be temperate based on the presence of a serine integrase, none of these phages encoded a known repressor typically required for lysogenic maintenance and conferral of superinfection immunity [10]. However, a lysogenic life cycle was supported by turbid plaque morphologies throughout isolation of our novel phages [37]. Using patch and liquid release assays, we determined that phage Powerpuff forms stable lysogens that release infective virions and is a temperate phage, (Fig 8A and 8B). Furthermore, immunity assays demonstrated that Cluster AZ phages Powerpuff, Lego, and YesChef were

**Fig 8. Phage Powerpuff forms stable lysogens that are immune to superinfection by related phages.** (A) Purified putative lysogens were streaked on a prepared lawn of *A. globiformis* B-2979. Zones of clearing were indicative of phage release. (B) Putative lysogens were grown in liquid culture then pelleted, after which supernatant was spot tested on *A. globiformis* B-2979 for titer. Plaque formations indicated spontaneous liquid release of phage particles and verified presence of stable lysogens. Two individual plaques were observed at the $1x10^{-9}$ dilution, giving a titer of ~$1x10^{12}$ PFU/mL of released phage. (C) 1:10 dilutions of *Arthrobacter* phage lysates were plated on WT *A. globiformis* B-2979 (left) and Powerpuff lysogens (right). Closely related phages were unable to lyse the Powerpuff lysogen, while unrelated phages retained infectivity with reduced efficiency.

unable to infect Powerpuff lysogens. The unrelated *Arthrobacter* Cluster FE phage Blue-Feather, Cluster AU phage Giantsbane, and Cluster AO phage Abba retained their infectivity with reduced efficiency of plaquing on the Powerpuff lysogens (Fig 8C, S4 Table). These results suggest that Clusters AZ and EH phages encode for currently unidentified repressors that convey homoimmunity [38].

## Discussion

The goal of this study was to describe the novel *A. globiformis* phages Powerpuff, Lego, and YesChef, as well as to characterize their relationships to phages infecting a variety of actinobacterial hosts. Previous large-scale studies of *Arthobacter* phages revealed minimal similarity to phages infecting non-*Arthrobacter* hosts [14]. However, more recent studies of *Microbacterium* phages have indicated similarities in both genome architecture and gene content between *Microbacterium* and *Arthrobacter* phages [10]. These findings confirm that such relationships do exist in the environment and suggest that as we isolate new phages, we will continue to identify cross-host relationships involving *Arthrobacter* phages.

As the number of sequenced actinobacteriophages increases, so does our understanding of the relationships among them. It was previously thought that phages infecting a common bacterial host would be most likely to exhibit increased genomic similarity [7]. More recent studies have provided evidence of vast genomic diversity amongst phages infecting a common host [6], as well as instances in which phages infecting unique hosts display substantial genomic similarities [9]. At the nucleotide level, the most similar phages to Powerpuff, Lego, and Yes-Chef outside of Cluster AZ belonged to *Microbacterium* Clusters EH and EB, *Arthrobacter* Cluster AK, and *Streptomyces* Cluster BJ. We observed high OrthoANIu values for many pairwise comparisons between phages of unique clusters. The vast majority of such comparisons (excluding those between Clusters AZ and EH) exhibited coverage values below 5%. This indicates that while there is a widespread prevalence of well-conserved genomic features among many of the phages included in this study, such features comprise only a small portion of each genome and are unlikely to represent a significant phylogenetic relationship. These results support previous findings which stated that *Arthobacter* phages are unlikely to share significant sequence similarity with phages infecting other actinobacterial genera [14].

In general, phages Liebe and Maureen accounted for the least similarity in nucleotide comparisons both within Cluster AZ and between Clusters AZ and EH. These phages were as

different from the other Cluster AZ phages as the Cluster AZ phages are collectively different from Cluster EH phages. The nucleotide dissimilarity of phages Liebe and Maureen from the remainder of Cluster AZ, at a level approximately equivalent to the similarities between Clusters AZ and EH, further illustrates the diversity of Cluster AZ and the complexity of these phage relationships. While nucleotide similarities between phage clusters were minimal, amino acid comparisons yielded stronger alignments between almost all of the genomes analyzed. The increase in alignment strength at an equivalent amino acid word length indicated a history of synonymous substitutions and a distant evolutionary relationship among these phages [36].

Analyses of shared gene content further complemented nucleotide and amino acid comparisons. GCS values between *Arthrobacter* phages in Cluster AZ and *Microbacterium* phages in Cluster EH either approached or exceeded the clustering parameter that has been applied to *Microbacterium* phages [10] and provided evidence for the close relationship between these phages at the gene content level. Though these values exceeded the new gene-content-based clustering parameter, we do not suggest that phages in Clusters AZ and EH should be clustered together. It is important to note as a limitation to this work, and for all phage comparative genomics, that clustering parameters depend upon the available dataset of sequenced phages and do not reflect fundamental separation points between groups of phages [9]. As more novel phages are isolated it is expected that previously discrete clusters may become less well-separated, even among phages infecting unique actinobacterial hosts.

The original analysis of Microbacterium phage Zeta1847 by Jacobs-Sera *et*. *al* designated it as a singleton, given GCS values of only ~20% with Cluster EH phages Floof and Percival [10]. Our updated analysis revealed that Zeta1847 shared more than 35% GCS with phages in Cluster EH, indicating that Zeta1847 is less genomically isolated from clustered *Microbacterium* phages than was previously thought. This finding resulted in the placement of Zeta1847 into Cluster EH. As more phage genes are sequenced, pham assignments may change and reveal previously unidentified relationships between both novel and previously isolated phages. In this case, the close relationship between Zeta1847 and the rest of Cluster EH is evidenced functionally as well, given that these phages are the only isolated *Microbacterium* phages which are known to encode an integrase and which may be able to undergo lysogeny [10].

Phages in Clusters AZ and EH shared a conserved genome architecture, with a high degree of synteny in the left arm of the genome and a similar order of conserved phams in the right arm. While some Cluster AZ phages encoded endolysins in the right arm of the genome, others encoded endolysins in the left arm similarly to Cluster EH phages, evidencing variability in the similarity of genome architecture both within and outside of each cluster. Many genes in the right arm of these genomes were either orphams (no known gene homologs) or had no known function. It is possible that as additional gene functions are assigned, further functional similarities and synteny will be observed between Clusters AZ and EH. A SplitsTree network phylogeny of shared gene content supported the proximity of the relationships between Clusters AZ and EH, while also demonstrating the diversity which exists among *Arthrobacter* phages [6, 14].

Previous research has stated that some genes are thought to "travel together" when being exchanged between genomes, including tail genes and DNA replication genes [7]. The functional significance of the genes which were shared between Clusters AZ and EH, including a large number of vital DNA processing, structural, and virion assembly genes, suggested common and conserved biological features and behaviors. Despite sharing considerable gene content with *M. foliorum* Cluster EH phages, phages Powerpuff, Lego, and YesChef were unable to infect this host. This is not entirely unexpected, given that shared gene content does not guarantee an expanded host range. For instance, while the majority of Cluster A phages are

known to infect mycobacterial hosts, phages belonging to Subcluster A15 are known to only infect *Gordonia* sp. [39].

Surprisingly, phage Powerpuff was able to form stable lysogens, despite lacking an identifiable repressor gene typically required for maintenance of lysogeny and conferral of superinfection immunity. Additionally, these lysogens were immune to superinfection by phages Powerpuff, Lego, and YesChef. All other phages in Clusters AZ and EH also lacked repressor genes but encoded an integrase in the same pham as Powerpuff's serine integrase (except for Cluster EH phage Percival). Phages are thought to be constrained by the kinetics of DNA packaging [40], which limits "genomic real estate." This makes the long-term conservation of unused or non-functional genes, or their replacement with functional homologs unlikely. If the Cluster EH phages were indeed lytic, it would be unexpected for phage Percival to encode a functionally homologous integrase gene in a different pham than the rest of these phages. Moreover, Cluster EH phages exhibit a genome architecture that has been previously described as distinct from known temperate phages [10]. However, here we find that these Cluster EH phages do in fact share a similar genome architecture with the Cluster AZ phages, including phage Powerpuff which is a temperate phage. This suggests that all Cluster AZ phages studied, as well as the Cluster EH phages, could be temperate phages with a yet unidentified repressor. If true, this would make the Cluster EH phages the first identified *Microbacterium* phages which are able to undergo lysogeny [10]. Since the lifecycle of only phage Powerpuff was assessed in the wet-lab, further experiments investigating the ability of Cluster EH phages to form stable lysogens would be necessary to confirm their bioinformatically predicted life cycles.

In summary, this research describes another case in which phages infecting different hosts share considerable genomic and biological similarities. We demonstrate, for the first time, significant conserved genomic content between *Arthrobacter* phages of Cluster AZ and phages of another actinobacterial host–particularly those belonging to Cluster EH, which infect *Microbacterium*. We also present the first experimental verification of lysogeny for a Cluster AZ phage, and suggest that closely related phages lacking known repressors may in fact be temperate. While the phage "puzzle" certainly remains incomplete, our findings serve to further illustrate the complexity of phage taxonomy and contribute to our understanding of actinobacteriophages and the characteristics which define them.

## Supporting information

**S1 Fig. Powerpuff, Lego, and YesChef have highly similar genomes.** Genomes were downloaded from Phamerator and formatted using Inkscape 1.0. Genes were sorted by general function or type, as indicated in the legend above. Powerpuff, Lego, and YesChef have highly similar genomes, with pairwise BLASTn scores of over 98.63% identity with at least 99% coverage and E-values of 0. Each genome was found to be between 43,446 and 44,651 bp in length, encoding between 69 and 71 genes. Notable dissimilarities included a gene duplication in phage Powerpuff (Powerpuff_29 and Powerpuff_31), which twice encoded a gene of unknown function that was present only once in phages Lego and YesChef. Phage Lego was found to encode a gene of unknown function (Lego_56) not found in Powerpuff or YesChef, located directly upstream of the gene encoding an endolysin. Phages Powerpuff and YesChef also encoded a gene of unknown function (Powerpuff_44 and YesChef_42) not found in phage Lego. This gene was positioned within a cassette of DNA processing genes in these phages. (TIF)

**S1 Table. NCBI BLASTn results for Powerpuff, Lego, YesChef, Tbone, and Kaylissa.** E-values for all comparisons were 0.0. (XLSX)

**S2 Table. PhagesDB BLASTn scores for Powerpuff query.**
(XLSX)

**S3 Table. Actinobacteriophage cluster information.** As of Nov. 2021.
(XLSX)

**S4 Table. Efficiencies of Plating (EOP).** EOP for phages spot titered on Powerpuff lysogens, relative to WT *A. globiformis* B-2979.
(XLSX)

## Acknowledgments

We would like to thank the Microbiology, Immunology, and Molecular Genetics Department, and the Dean of Life Sciences Division at UCLA for programmatic support. We thank Esther Choe and Sophia Wang for their assistance in phage isolation, genome annotation, and preliminary analysis. We also thank Rebecca A. Garlena and Daniel A. Russell at the Pittsburgh Bacteriophage Institute for genome sequence and assembly and Travis Mavrich, Welkin Pope, Debbie Jacobs-Sera, and Graham Hatfull with the HHMI Science Education Alliance-Phage Hunters Advancing Genomics and Evolutionary Science (SEA-PHAGES) program for programmatic support. We would also like to thank Daniel A. Russell and Graham Hatfull for clustering assistance and manuscript feedback. The authors acknowledge the use of instruments at the Electron Imaging Center for NanoMachines supported by NIH (1S10RR23057 to ZHZ) and CNSI at UCLA. The authors also acknowledge the Agricultural Research Service (ARS) Culture Collection for providing *Microbacterium foliorum* NRRL B-24224.

## Author Contributions

**Conceptualization:** Andrew Kapinos, Amanda C. Freise, Jordan Moberg Parker.

**Data curation:** Amanda C. Freise, Jordan Moberg Parker.

**Formal analysis:** Andrew Kapinos, Pauline Aghamalian, Erika Capehart, Anya Alag, Heather Angel, Eddie Briseno, Byron Corado Perez, Emily Farag, Hilory Foster, Abbas Hakim, Daisy Hernandez-Casas, Calvin Huang, Derek Lam, Maya Mendez, Ashley Min, Nikki Nguyen, Alexa L. Omholt, Emily Ortiz, Lizbeth Shelly Saldivar, Jack Arthur Shannon, Rachel Smith, Mihika V. Sridhar, An Ta, Malavika C. Theophilus, Ryan Ngo, Canela Torres.

**Investigation:** Andrew Kapinos, Pauline Aghamalian, Erika Capehart, Anya Alag, Heather Angel, Eddie Briseno, Byron Corado Perez, Emily Farag, Hilory Foster, Abbas Hakim, Daisy Hernandez-Casas, Calvin Huang, Derek Lam, Maya Mendez, Ashley Min, Nikki Nguyen, Alexa L. Omholt, Emily Ortiz, Lizbeth Shelly Saldivar, Jack Arthur Shannon, Rachel Smith, Mihika V. Sridhar, An Ta, Malavika C. Theophilus, Ryan Ngo, Canela Torres.

**Methodology:** Andrew Kapinos, Krisanavane Reddi, Amanda C. Freise.

**Project administration:** Krisanavane Reddi, Amanda C. Freise, Jordan Moberg Parker.

**Resources:** Krisanavane Reddi.

**Supervision:** Amanda C. Freise, Jordan Moberg Parker.

**Visualization:** Andrew Kapinos, Pauline Aghamalian, Erika Capehart, Anya Alag, Heather Angel, Eddie Briseno, Byron Corado Perez, Emily Farag, Hilory Foster, Abbas Hakim, Daisy Hernandez-Casas, Calvin Huang, Derek Lam, Maya Mendez, Ashley Min, Nikki Nguyen, Alexa L. Omholt, Emily Ortiz, Lizbeth Shelly Saldivar, Jack Arthur Shannon, Rachel Smith, Mihika V. Sridhar, An Ta, Malavika C. Theophilus, Ryan Ngo, Canela Torres.

**Writing – original draft:** Andrew Kapinos, Pauline Aghamalian, Erika Capehart, Anya Alag, Heather Angel, Eddie Briseno, Byron Corado Perez, Emily Farag, Hilory Foster, Abbas Hakim, Daisy Hernandez-Casas, Calvin Huang, Derek Lam, Maya Mendez, Ashley Min, Nikki Nguyen, Alexa L. Omholt, Emily Ortiz, Lizbeth Shelly Saldivar, Jack Arthur Shannon, Rachel Smith, Mihika V. Sridhar, An Ta, Malavika C. Theophilus.

**Writing – review & editing:** Andrew Kapinos, Krisanavane Reddi, Amanda C. Freise, Jordan Moberg Parker.

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
