## [Decision Letter · Decision Letter 0]

20 Oct 2021

PONE-D-21-30582Novel Cluster AZ *Arthrobacter* phages Powerpuff, Lego, and YesChef exhibit close functional relationships with *Microbacterium* phagesPLOS ONE

Dear Dr. Jordan Moberg Parker,

Thank you for submitting your manuscript to PLOS ONE. After careful consideration, we feel that it has merit but does not fully meet PLOS ONE’s publication criteria as it currently stands. Therefore, we invite you to submit a revised version of the manuscript that addresses the points raised during the review process.

We look forward to receiving your revised manuscript.

Kind regards,

Islam Hamim, PhD

Academic Editor

PLOS ONE

Journal Requirements:

Additional Editor Comments:

Minor revision required.

Reviewers' comments:

Reviewer's Responses to Questions

**Comments to the Author**

1. Is the manuscript technically sound, and do the data support the conclusions?

Reviewer #1: Yes

Reviewer #2: Yes

Reviewer #3: Yes

Reviewer #4: Yes

2. Has the statistical analysis been performed appropriately and rigorously? 

Reviewer #1: N/A

Reviewer #2: Yes

Reviewer #3: N/A

Reviewer #4: Yes

3. Have the authors made all data underlying the findings in their manuscript fully available?

Reviewer #1: Yes

Reviewer #2: Yes

Reviewer #3: Yes

Reviewer #4: Yes

4. Is the manuscript presented in an intelligible fashion and written in standard English?

Reviewer #1: No

Reviewer #2: Yes

Reviewer #3: Yes

Reviewer #4: Yes

5. Review Comments to the Author

Reviewer #1: The manuscript is very well written. Here the authors have successfully detected and characterize novel Cluster AZ Arthrobacter phages Powerpuff, Lego, and YesChef that exhibit some relationships at functional level with other phages specific for Microbacterium.

In this study detailed genetic characterization at molecular were done that supports the claimed fining. Many of these data are already available in GenBank for public use. Most of the methodologies are well described and reproduceable. Finding of this study will certainly enhance our knowledge for better understand of novel phage biology at molecular level.

However few points need to be addressed as follows:

Introduction:

Line 88 to 110 describe the finds of this study. It doesn’t make any sense to write findings/results under introduction. Take these out from introduction, instead, please state clearly the aim of this study here.

Methodology:

Line 163. Please expand the host range assay.

Results:

Line 267, please define what dose this “weak alignment” refers here??

Discussion:

Line 441. At the end please add examples of those few other host genera??

Add the weakness of this study.

Reviewer #2: Although this article uses mostly bioinformatics for analyzing sequences, it does not fall into the descriptive and speculative field. Indeed, the manuscript presents wet data that add value to interpretation and leads to important final conclusions.

Minor corrections and suggestions:

The writing and language usage is correct and orthodox, however, there are many instances where reader gets breathless. There are many instances where a colon or a semicolon can be the place to cut a sentence (i.g. semicolon at L37P2 in the abstract section). Indeed, it is easy to find sentences spanning 7-10 lines. This reviewer thing it is necessary to cut the length of several sentences.

The complexity of presenting scientific data is more of a problem that a virtue. The manuscript quickly run into the topic of bacteriophage clusters with designed names that will be unfamiliar, unless reader is really into the Arthrobacter world and Arthrobacter phages. To easy the understanding, and because there are few restrictions in the number of figures and tables, this reviewer suggests a simple table with the names of phage clusters, the hosts, number of phages and any relevant information (presence of lytic, temperate phages may be informative).

Final comments. If the authors address the minor corrections and comments presented above, this reviewer agrees that the manuscript is suitable for publication in PlosOne.

Reviewer #3: The manuscript entitled,'Novel Cluster AZ Arthrobacter phages Powerpuff, Lego, and YesChef exhibit close functional relationships with Microbacterium phages' is technically sound with all necessary information.

Reviewer #4: In Fig. 7;

It is suggested to put the bootstrap value(s) to closely perceive the intra-cluster similarities in that phylogenic tree.

In Fig.8. Plate B;

Powerpuff lysogen titer spot was identified with weak signal (no. 7-9). But it was not discussed well either those are individual copies or mutant of Powerpuff phages.

6. PLOS authors have the option to publish the peer review history of their article (what does this mean?). If published, this will include your full peer review and any attached files.

Reviewer #1: **Yes: **Md. Tanvir Rahman

Reviewer #2: **Yes: **Alejandro A. Hidalgo

Reviewer #3: No

Reviewer #4: **Yes: **Dr. Mohammad Delwar Hossain, Professor of Plant Pathology, Faculty of Agriculture, Bangladesh Agricultural University, Mymensingh-2202.

---

## [Author Response · Author response to Decision Letter 0]

3 Dec 2021

Reviewer #1: 

"Introduction:

Line 88 to 110 describe the finds of this study. It doesn’t make any sense to write findings/results under introduction. Take these out from introduction, instead, please state clearly the aim of this study here."

Thank you for the feedback, we have revised the last paragraph of the introduction to state the goals of the paper more clearly. In constructing this paper, we followed the structure described in Mensh B, Kording K (2017) “Ten simple rules for structuring papers,” PLOS Computational Biology 13(9): e1005619. https://doi.org/10.1371/journal.pcbi.1005619. This editorial recommends that the summary of the introduction should include the approach and brief results. We have however significantly condensed this section. 

"Methodology:

Line 163. Please expand the host range assay."

The host range assay methodology has been expanded with additional detail.

"Results:

Line 267, please define what does this “weak alignment” refers here??"

Section has been revised.

"Discussion:

Line 441. At the end please add examples of those few other host genera??

Add the weakness of this study."

Line clarified to reflect that we are referring to actinobacterial genera. Line added on the weakness of comparative genomics studies being that clustering parameters rely on limited available data. Line added acknowledging lifecycle conclusions were based on wetlab data from a single phage.

Reviewer #2: 

"Minor corrections and suggestions:

The writing and language usage is correct and orthodox, however, there are many instances where reader gets breathless. There are many instances where a colon or a semicolon can be the place to cut a sentence (i.g. semicolon at L37P2 in the abstract section). Indeed, it is easy to find sentences spanning 7-10 lines. This reviewer thing it is necessary to cut the length of several sentences."

Thank you for the constructive feedback. The manuscript has been edited throughout to reduce verbosity and improve clarity.

"The complexity of presenting scientific data is more of a problem that a virtue. The manuscript quickly run into the topic of bacteriophage clusters with designed names that will be unfamiliar, unless reader is really into the Arthrobacter world and Arthrobacter phages. To easy the understanding, and because there are few restrictions in the number of figures and tables, this reviewer suggests a simple table with the names of phage clusters, the hosts, number of phages and any relevant information (presence of lytic, temperate phages may be informative)."

A supplemental table (S4 Table) has been added with summary information of actinobacteriophage cluster data.

Reviewer #4: 

"In Fig. 7; It is suggested to put the bootstrap value(s) to closely perceive the intra-cluster similarities in that phylogenic tree."

SplitsTree network phylogenies are different from phylogenetic trees in that they typically do not have bootstrap values. In a classic phylogenetic tree, relationships are predicted with a given confidence value based on sequence (dis)similarities. With a network phylogeny, SplitsTree uses the presence or absence of each gene to construct a "network" that represents such differences visually. This is less prediction-based, and thus confidence values are not required (for instance, works by other phage scientists do not present bootstrap values on these types of trees). Lines were added to the comparative genomic analysis section of the methods and the results section for figure 7 to address the potential confusion.

"In Fig.8. Plate B; Powerpuff lysogen titer spot was identified with weak signal (no. 7-9). But it was not discussed well either those are individual copies or mutant of Powerpuff phages."

In this experiment, spot tittering was performed to get a rough estimate of the number of phages released from the lysogen cells. For each spot the presence of clearings indicates phage in that dilution and a rough titer can be calculated since we see two individual plaques at 1e-9. We have no way of determining if released phages are mutants from this assay. The methods section and figure legend for this experiment have been revised for clarity.

---

## [Decision Letter · Decision Letter 1]

29 Dec 2021

Novel Cluster AZ *Arthrobacter* phages Powerpuff, Lego, and YesChef exhibit close functional relationships with *Microbacterium* phages

PONE-D-21-30582R1

Dear Dr. Jordan Moberg Parker,

We’re pleased to inform you that your manuscript has been judged scientifically suitable for publication and will be formally accepted for publication once it meets all outstanding technical requirements.

Kind regards,

Islam Hamim, PhD

Academic Editor

PLOS ONE

Additional Editor Comments (optional):

Accepted

Reviewers' comments:

Reviewer's Responses to Questions

**Comments to the Author**

1. If the authors have adequately addressed your comments raised in a previous round of review and you feel that this manuscript is now acceptable for publication, you may indicate that here to bypass the “Comments to the Author” section, enter your conflict of interest statement in the “Confidential to Editor” section, and submit your "Accept" recommendation.

Reviewer #2: All comments have been addressed

2. Is the manuscript technically sound, and do the data support the conclusions?

Reviewer #2: Yes

3. Has the statistical analysis been performed appropriately and rigorously? 

Reviewer #2: Yes

4. Have the authors made all data underlying the findings in their manuscript fully available?

Reviewer #2: Yes

5. Is the manuscript presented in an intelligible fashion and written in standard English?

Reviewer #2: Yes

6. Review Comments to the Author

Reviewer #2: The authors have addressed the suggestions in a positive way. This reviewer has no further comments and accept the manuscript in this revised version.

7. PLOS authors have the option to publish the peer review history of their article (what does this mean?). If published, this will include your full peer review and any attached files.

Reviewer #2: No

---

## [Editor Report · Acceptance letter]

4 Jan 2022

PONE-D-21-30582R1 

Novel Cluster AZ *Arthrobacter* phages Powerpuff, Lego, and YesChef exhibit close functional relationships with *Microbacterium* phages 

Dear Dr. Moberg Parker:

I'm pleased to inform you that your manuscript has been deemed suitable for publication in PLOS ONE. Congratulations! Your manuscript is now with our production department. 

Kind regards, 

on behalf of

Dr. Islam Hamim 

Academic Editor

PLOS ONE